# Detection of genetic divergence among some wheat (*Triticum aestivum* L.) genotypes using molecular and biochemical indicators under salinity stress

**Salah M. H. Gowayed[1,2], Diaa Abd El-Moneim[3]** *

**1** Department of Biology, College of Science, University of Jeddah, Jeddah, Saudi Arabia, **2** Department of Botany, Faculty of Agriculture, Suez Canal University, Ismailia, Egypt, **3** Department of Plant Production, (Genetic Branch), Faculty of Environmental and Agricultural Sciences, Arish University, El- Arish, Egypt

* dabdelmoniem@Aru.edu.eg

## Abstract

Wheat has remarkable importance among cereals in Egypt. Salt stress affects plant growth, development, and crop productivity. Therefore, salinity tolerance is an essential trait that must be incorporated in crops. This research aimed to investigate molecular and biochemical indicators and defence responses in seedlings of 14 Egyptian wheat genotypes to distinguish the most contrasting salt-responsive genotypes. Analysis of ISSR and SCoT markers revealed high polymorphism and reproducible fingerprinting profiles for evaluating genetic variability within the studied genotypes. The HB-10 and SCoT 1 primers had the highest values for all the studied parameters. All the tested primers generated a set of 66 polymorphic bands among tolerant and sensitive genotypes. The transcript profiles of eight *TaWRKY* genes showed significant induction under the salinity treatments. Moreover, the expression of *TaWRKY6* for genotypes Sids 14 and Sakha 93 sharply increased and recorded the highest expression, while the expression of *TaWRKY20* for Misr 1 recorded the lowest expression. Under salt stress, the total sugar, proline, and phenolic contents increased significantly, while the chlorophyll content decreased significantly. Additionally, five peroxidase and polyphenol oxidase isoforms were observed in treated leaves and clustered into five different patterns. Some isoforms increased significantly as salinity levels increased. This increase was clearer in salt-tolerant than in salt-sensitive genotypes. Eighteen protein bands appeared, most of which were not affected by salinity compared with the control, and specific bands were rare. Generally, the Sids 14, Sakha 93, Sohag 4, and Gemmeiza 12 genotypes are considered salt tolerant in comparison to the other genotypes.

## 1 Introduction

Wheat (*Triticum aestivum* L.) is one of the most important cereal crops worldwide, especially in Egypt. Therefore, compensation for the gap between wheat production and consumption,

**Data Availability Statement:** All relevant data are within the manuscript and its Supporting Information files.

**Funding:** This work was funded by the University of Jeddah, Saudi Arabia, under grant no. (UJ-02-016-DR). The authors, therefore, acknowledge with thanks the University technical and financial support.

**Competing interests:** The authors have declared that no competing interests exist.

especially under many environmental pressures, is very important [1]. Salinity is a fundamental limiting factor for crop production in dry and irrigated fields [2]. Salinity tolerance in plants is not a fixed trait and exhibits significant variability within plants, particularly among species and the stage of growth [3]. Evaluation of the germplasm's genetic diversity is considerably important to facilitate the selection of genotypes with higher diversity and superior performance under certain conditions. Inter-simple sequence repeats (ISSR) and start codon targeted (SCoT) markers are efficiently used for genetic variability assessment of plants and may aid as markers of different traits such as salinity tolerance. In this regard, Kaushiik *et al.* found four ISSR amplified bands in F3 plants, and such polymorphic bands demonstrated greater chances of linkage with the genes/QTLs for salinity tolerance [4]. The ISSR marker is simple and reproducible and requires small amounts of DNA without previous sequence information [5]. ISSR primers are designed from SSR motifs and can be used for any plant species containing a sufficient number and distribution of SSR motifs in the genome [6]. Thus, ISSRs are vastly used in genetic diversity studies in wheat [7], barley [8], and beans [9]. The SCoT marker is a dominant and reproducible marker based on the short conserved region flanking the ATG start codon in plant genes [10]. This type of marker could have an essential role in genotyping and detecting polymorphisms in wheat [11]. Gene expression is regulated by transcription factors (TFs), which provide plants with multiple mechanisms for countering biotic and abiotic stresses and modulating developmental processes [12]. WRKY TFs, one of the ten largest transcription factor families, are described by a highly conserved WRKYGQK heptapeptide at the N-terminus and a zinc finger-like motif at the C-terminus [13]. In wheat, many researchers have uncovered the fundamental role of *TaWRKYs* in plant molecular salinity responses. In this regard, overexpression of *TaWRKY 2&9* and *TaWRKY10* led to improved drought and salt adaptation in transgenic plants [14–16]. On the other hand, it is well recognized that salinity stress induces stomatal closure and consequently reduces $CO_2$ fixation [17]. Therefore, an over-reduction of the photosynthetic electron chain can occur, which triggers the production of reactive oxygen species [18], leading to oxidative stress. Notably, higher plants develop various protective mechanisms to diminish oxidative damage stimulated by salinity stress. Furthermore, the synthesis and accumulation of soluble sugar, proline, and phenolic compounds are stimulated in response to biotic and abiotic stresses and are considered to be essential indices of the salt tolerance ability of plants. Additionally, the chlorophyll and total carotenoid contents are reduced in wheat leaves under salt conditions [19]. Likewise, considerable evidence reported by [20, 21] and [22] has shown significant increases in the sugar, proline, and total phenolic contents as a consequence of salinity treatment in wheat. In the same context, plants produce a number of antioxidant enzymes that protect their cells from potential cytotoxic effects under stressful environments [23]. El-Beltagi *et al.* [24] and [25] demonstrated that salt stress induced the activity of peroxidase (PX) and polyphenol oxidase (PPO) enzymes and increased the detected number and intensity of isozyme bands. Moreover, protein accumulation may act as an energy reservoir or control the osmotic potential in plants exposed to salinity [26]. Furthermore, Khalipe *et al.* [27] stated that increased protein content after NaCl treatment led to an increase in the tolerance mechanisms towards NaCl salinity of wheat varieties. Combining DNA markers, gene expression, and biochemical analysis is required to fully understand the influence of high salinity in wheat and is considered to be a promising strategy for elucidating the plant stress response mechanism. As a consequence, the overall objective of this study was to characterize some Egyptian wheat genotypes that differ in their response to salinity stress using diverse indicators, including molecular and biochemical factors. Molecular aspects include the estimation of SCoTs and ISSRs in assessing the nature and extent of genetic diversity and studying the expression levels of some *TaWRKY* genes under different concentrations of NaCl, while biochemical aspects include determining the changes in total sugar,

proline, total phenolic, and chlorophyll contents, PX & PPO isozymes, and protein patterns in wheat leaves under salinity stress.

## 2 Materials and methods

### 2.1 Plant materials and growth conditions

Fourteen Egyptian wheat (*Triticum aestivum)* genotypes, namely, Giza 168, Misr 1, Misr 2, Misr 3, Shandaweel 1, Sids 1, Sids 12, Sids 14, Sakha 93, Sakha 95, Bani Seuf 7, Sohag 4, Sohag 5, and Gemmeiza 12, were used in the experiments. Seeds were sterilized and then germinated for three weeks using the method described by [28]. In the fourth week, salinity stress treatments (0, 50, 150, and 250 mM NaCl) were dissolved in distilled water. One week later, leaves were immediately harvested, frozen in liquid nitrogen, and kept at −80˚C for further analysis.

### 2.2 ISSR and SCoT marker analysis

**2.2.1 Genomic DNA extraction and PCR procedures.**   Seeds were cultivated into pots containing peat moss under greenhouse conditions at 22˚C. The total genomic DNA was extracted from young leaf pieces using a DNA Plant Kit (Qiagen). Five primers/markers (S1 Table) were screened for the studied genotypes. SCoT and ISSR amplification were performed as described by [10] and [29], respectively. PCR products were visualized by conventional agarose gel electrophoresis.

**2.2.2 DNA banding pattern and markers efficiency analysis.**   The clear and unambiguous DNA banding patterns generated from ISSR and SCoT markers were considered for study and analysed by Gel Works ID advanced software. The presence or absence of each recorded band for each genotype is indicated by (1) or (0). To assess the informativeness of the studied markers, the following parameters were calculated for each primer: polymorphism information content (PIC) [30], effective multiplex ratio (EMR) [31], marker index (MI) [31], and resolving power (Rp) [32]. GenAlEx software version 6.5 [33] was used to calculate the average number of alleles per loci (Na), average number of effective alleles per loci (Ne), percentage of polymorphic loci (P), expected heterozygosity (He), and unbiased expected heterozygosity (uHe) for each primer across the studied genotypes based on the frequency of alleles of each locus. Genetic similarity was calculated by Jaccard's coefficient. A dendrogram was generated with the unweighted pair group method with the arithmetic mean (UPGMA) algorithm using the computational package MVSP V. 3.1.

### 2.3 RNA extraction and cDNA synthesis

RNA was extracted using a TRIzol® kit (Invitrogen Inc.) from the leaves and then converted into cDNA with the high capacity cDNA Reverse Transcription Kit (Applied Biosystems). All experiments were repeated three times. *TaActin* was used as a housekeeping gene. The studied genes and their corresponding primers are shown in (S2 Table).

### 2.4 Biochemical analysis

**2.4.1 Determination of photosynthetic pigments.**   For each fresh leaf sample, 0.2 g was homogenized in 10 mL of 96% methanol in the dark at four ˚C. The extract was centrifuged at $2500 \times g$ for 10 min, and then the supernatant was diluted four-fold with methanol. The absorbance of the methanol-diluted supernatant was estimated at 666, 653, and 470 nm for Chl a, Chl b, and carotenoids, respectively, using a spectrophotometer. The total content of each parameter was estimated by the formula reported by [34].

**2.4.2 Measurement of the total sugar, proline, and total phenolic contents.** At the end of the salinity experiments, 0.5 g of fresh weight (FW) was collected for treated and non-treated samples. The total sugar, proline, and total phenolic contents were estimated according to the methods described by [35–37].

**2.4.3 Isozyme analysis.** Native polyacrylamide gel electrophoresis was performed according to [38]. To distinguish isozyme variability between normal and treated plants, two isozyme systems were used: peroxidase and polyphenol oxidase. Subsequently, for peroxidase, gels were incubated in the dark at room temperature until bands appeared in 0.125 g of benzidine·2HCl and 2 ml of glacial acetic acid and completed with distilled water up to 50 ml. Then, five drops of hydrogen peroxidase were added [39]. The polyphenol oxidase gel was incubated at 30˚C for 30 min until bands appeared in 100 ml of 0.1 M sodium phosphate buffer, 15 mg of catechol, and 50 mg of $C_6H_7NO_3S$.

**2.4.4 Protein analysis.** Protein banding patterns were analysed by sodium dodecyl sulphate polyacrylamide gel electrophoresis (SDS-PAGE). The total protein was extracted according to [40]. Leaves of wheat samples collected from the treated and non-treated plants were used for total protein extraction. The protein content was determined according to the method of [41]. The presence or absence of each recorded protein band for each genotype/treatment was indicated by (1) or (0).

## 2.5 Statistical analysis

All data are represented as the mean ± SD of three replicates. One- and two-way analysis of variance (ANOVA) were used to test the hypothesis that the genotype and concentration of salinity affect the studied characteristics of plants. If there were significant differences between the means, Tukey's post hoc comparisons among different groups were performed. *P* values ≤ *0.05* were considered to be statistically significant for all statistical tests. Data and statistical analysis were carried out using Excel 2016 and Minitab V.19.

**2.5.1 Multivariate analysis.** Principal coordinate analysis PCoA [42] was used to find the eigenvalues and eigenvectors of a matrix containing the similarities among all genotypes based on their gene expression patterns. The scatter plot shows all data points (rows) plotted in the coordinate system given by the PCoA. Canonical correspondence analysis (CCA) [43] was used for the correspondence analysis of genotype/gene expression and the ISSR and SCoT band matrix. PAST software V4.01 was used in the canonical correspondence analysis [44].

# 3 Results

## 3.1 ISSR and SCoT polymorphism assessment

The level of polymorphism and a comparison of the discriminating capacity of ISSR and SCoT markers are summarized in (Table 1). Using five ISSR primers, a total of 77 scorable amplified bands (66 polymorphic and 11 monomorphic) were observed. Their molecular size ranged from 185 to 1970 bp (S1A Fig and Table 1). The HB-10 primer recorded the highest value of polymorphism (P %) at 96%. Moreover, several parameters were computed to describe the informativeness of the studied primers. The HB-10 primer recorded the highest values of 23.04, 18.15, 10.61, 1.96, 0.19, and 0.20 for EMR, MI, Rp, Na, He, and uHe, respectively, whereas HB-12 and 49 A recorded the highest values for PIC (0.87) and Ne (1.33). In addition, the patterns of primers 49A, HB-08, and HB-13 revealed the lowest values for all studied parameters. On the other hand, using five primers of SCoT revealed 78 amplified fragments with molecular sizes ranging from 185 to 2330 bp, whereas 64 fragments were polymorphic, and the residual 14 fragments were monomorphic (S1B Fig and Table 1). The SCoT 1 Primer recorded the highest value of polymorphism (P %) at 92.3%. The SCoT 1 Primer recorded the

**Table 1. Number and types of amplified DNA bands, levels of polymorphism, and comparison of informativeness obtained with ISSR and SCoT markers in 14 wheat genotypes.**

|  | ISSR | | | | | | SCoT | | | | | |
|---|---|---|---|---|---|---|---|---|---|---|---|---|
|  | 49A | HB-08 | HB-10 | HB-12 | HB-13 | Ave | SCoT 1 | SCoT 2 | SCoT 3 | SCoT 4 | SCoT 8 | Ave. |
| MB | 3 | 3 | 1 | 1 | 3 | 2.20 | 2 | 2 | 3 | 5 | 2 | 2.80 |
| UB | 3 | 6 | 11 | 11 | 5 | 7.20 | 3 | 7 | 6 | 3 | 7 | 5.20 |
| PB | 8 | 9 | 24 | 17 | 8 | 13.2 | 24 | 10 | 8 | 9 | 13 | 12.80 |
| TAB | 11 | 12 | 25 | 18 | 11 | 15.4 | 26 | 12 | 11 | 14 | 15 | 15.60 |
| FS | 200–770 | 325–1250 | 210–1970 | 290–960 | 185–700 |  | 160–2330 | 275–1180 | 145–980 | 145–880 | 185–1290 |  |
| PIC | 0.53 | 0.71 | 0.79 | 0.87 | 0.53 | 0.69 | 0.76 | 0.75 | 0.65 | 0.60 | 0.77 | 0.71 |
| EMR | 5.82 | 6.75 | 23.04 | 16.06 | 5.82 | 11.5 | 22.15 | 8.33 | 5.82 | 5.79 | 11.27 | 10.67 |
| MI | 3.11 | 4.82 | 18.15 | 13.98 | 3.11 | 8.63 | 16.84 | 6.21 | 3.80 | 3.44 | 8.69 | 7.80 |
| P% | 72.7 | 75 | 96 | 94.4 | 72.7 | 0.82 | 92.3 | 83.3 | 72.7 | 64.3 | 86.7 | 79.9 |
| Rp | 10.24 | 6.85 | 10.61 | 4.65 | 10.25 | 8.52 | 13.82 | 6.11 | 7.62 | 11.34 | 6.86 | 9.15 |
| Na | 1.72 | 1.75 | 1.96 | 1.94 | 1.72 | 1.81 | 1.92 | 1.83 | 1.72 | 1.64 | 1.86 | 1.79 |
| Ne | 1.33 | 1.14 | 1.31 | 1.20 | 1.27 | 1.25 | 1.42 | 1.24 | 1.14 | 1.17 | 1.24 | 1.24 |
| He | 0.19 | 0.10 | 0.19 | 0.13 | 0.15 | 0.15 | 0.25 | 0.15 | 0.09 | 0.12 | 0.16 | 0.15 |
| uHe | 0.20 | 0.10 | 0.20 | 0.14 | 0.16 | 0.16 | 0.26 | 0.15 | 0.10 | 0.12 | 0.16 | 0.15 |

**MB**: monomorphic band, **UB**: unique band, **PB**: polymorphic band, **TAB**: total amplified bands, **FS**: fragment size, **PIC**: polymorphic information content, **EMR**: effective multiplex ratio, **MI**: marker index, **P%**: percent of polymorphism, **Rp**: resolving power, **Na**: average number of alleles per loci, **Ne**: average number of effective alleles per loci, **He**: expected heterozygosity, **uHe**: unbiased expected heterozygosity.

highest values of EMR (22.15), MI (16.84), Rp (13.82), Na (1.92), Ne (1.42), He (0.25), and uHe (0.26), while, SCoT 8 recorded the highest value for PIC (0.77). In contrast, primers SCoT 4 & 3 showed the lowest values. The results showed that the HB-10 and SCoT 1 primers were the most informative and had greater potential among the primers used in this study. Interestingly, the number of positive and negative unique bands was 66 bands, scored across studied genotypes using both ISSR and SCoT primers. Forty amplified bands (36 positive and four negative) were derived from ISSR primers, whereas 26 bands were derived from SCoT primers (Table 2). Primer HB-10 revealed the highest number of unique bands (13 markers), while the SCoT 2 and SCoT 8 primers showed the highest number of unique bands (seven markers). Moreover, the studied genotypes varied considerably in their detected unique markers. Shandaweel 1 had the highest number of unique bands (14 markers revealed by SCoT and three markers revealed by ISSR). Conversely, the lowest number of unique bands was scored by the Misr 1 genotype (one marker).

The similarity coefficient among the studied genotypes using ISSR and SCoT combined data varied from (0) between Sakha 93 and Bani Seuf 7 to (1) between Misr 3 and Giza168 (S3 Table). Cluster analysis for ISSR and SCoT combined data revealed two main clusters (S2 Fig). The first cluster contained only the Shandaweel 1 genotype, while the second cluster was divided into two subclusters. The first subcluster included two genotypes (Giza 186 and Misr 1). Finally, the other subcluster included the rest of the studied genotypes. Interestingly, the phylogenetic tree for the SCoT data and the combined data were exactly the same.

### 3.2 Expression analysis of *TaWRKY* under NaCl treatment

Since WRKY genes are predominantly related to plant defence responses, changes in the expressions of eight *TaWRKY* genes in response to salinity stress were investigated. ANOVA for mRNA transcript expression revealed that the main effects of genotypes, salinity

**Table 2. The number of positive and negative unique ISSR and SCoT markers recorded in 14 wheat genotypes.**

| | Negative unique ISSR markers | | | Positive unique ISSR markers | | | Negative unique SCoT markers | | |
|---|---|---|---|---|---|---|---|---|---|
| | Primer | Size of band (bp) | Total | Primer | Size of band (bp) | Total | Primer | Size of band (bp) | Total |
| **Misr 2** | 49A | 660 | 4 | HB-10 | 300/355 | 2 | SCoT 3 | 370 | 1 |
| | HB-10 | 280/440 | | | | | | | |
| | HB-13 | 700 | | | | | | | |
| **Misr 3** | HB-10 | 455 | 2 | - | - | - | - | - | - |
| | HB-12 | 395 | | | | | | | |
| **Sids 1** | HB-10 | 850 | 2 | - | - | - | - | - | - |
| | HB-13 | 655 | | | | | | | |
| **Sids 12** | HB-12 | 510 | 3 | - | - | - | SCoT 8 | 525 | 1 |
| | HB-13 | 185/205 | | | | | | | |
| **Sids 14** | 49A | 775 | 4 | - | - | - | SCoT 2 | 560 | 3 |
| | HB-08 | 830 | | | | | SCoT 8 | 490/616 | |
| | HB-10 | 1260 | | | | | | | |
| | HB-12 | 590 | | | | | | | |
| **Sakha 93** | 49A | 690 | 4 | - | - | - | - | - | - |
| | HB-10 | 935/953/1185 | | | | | | | |
| **Bani Seuf 7** | HB-10 | 1220 | 3 | - | - | - | - | - | - |
| | HB-12 | 700 | | | | | | | |
| | HB-13 | 210 | | | | | | | |
| **Sohag 4** | HB-10 | 1150 | 2 | 49A | 200 | 1 | SCoT 2 | 577 | 1 |
| | HB-12 | 580 | | | | | | | |
| **Sohag 5** | HB-12 | 600 | 1 | - | - | - | SCoT 8 | 1132 | 1 |
| **Shandaweel 1** | HB-08 | 780/1080 | 3 | - | - | - | SCoT 1 | 1280/1820 | 14 |
| | HB-12 | 675 | | | | | SCoT 2 | 665/835/1180 | |
| | | | | | | | SCoT 3 | 143/405/560/720/980 | |
| | | | | | | | SCoT 4 | 880 | |
| | | | | | | | SCoT 8 | 577/723/1290 | |
| **Sakha 95** | - | - | 0 | - | - | - | SCoT 2 | 275 | 1 |
| **Gemmeiza 12** | HB-08 | 890 | 2 | - | - | - | SCoT 4 | 143 | 1 |
| | HB-10 | 1600 | | | | | | | |
| **Giza 168** | HB-08 | 950/1250 | 3 | HB-13 | 513 | 1 | SCoT 1 | 930 | 3 |
| | HB-12 | 610/800/960 | | | | | SCoT 2 | 685 | |
| | | | | | | | SCoT 4 | 188 | |
| **Misr 1** | HB-12 | 515 | 1 | - | - | - | - | - | - |

treatments, and interactions were significant at $P \leq 0.05$, except the interaction was not significant for *TaWRKY2* expression. In the same trend, Tukey's test showed a significant difference within different salinity treatments for all the studied genotypes except the expression of Misr 3 in *TaWRKY2*, Sids 12 in *TaWRKY4*, and Sids 1 in *TaWRKY8*. Fig 1A and 1B showed that the mRNA levels of Sids 14 (*TaWRKY6*), Sakha 93 (*TaWRKY6*), and Sids 14 (*TaWRKY20*) recorded the highest transcript levels of 25-, 14.9-, and 3.5-fold at 50, 150, and 250 mM NaCl, respectively, compared with the untreated control. The data in (S4 Table) showed that all the studied genes were upregulated under different NaCl concentrations for most of the studied genotypes, while the number of upregulated genes at 50 and 150 mM NaCl was higher than that at 250 mM NaCl. In this regard, *TaWRKY4*, *8*, *6*, and *TaWRKY2*, *20* recorded the highest and lowest number of upregulated genotypes, respectively. Furthermore, (S5 Table) showed that Sakha 93 had the highest number of upregulated genes, (seven and eight genes at 50 and

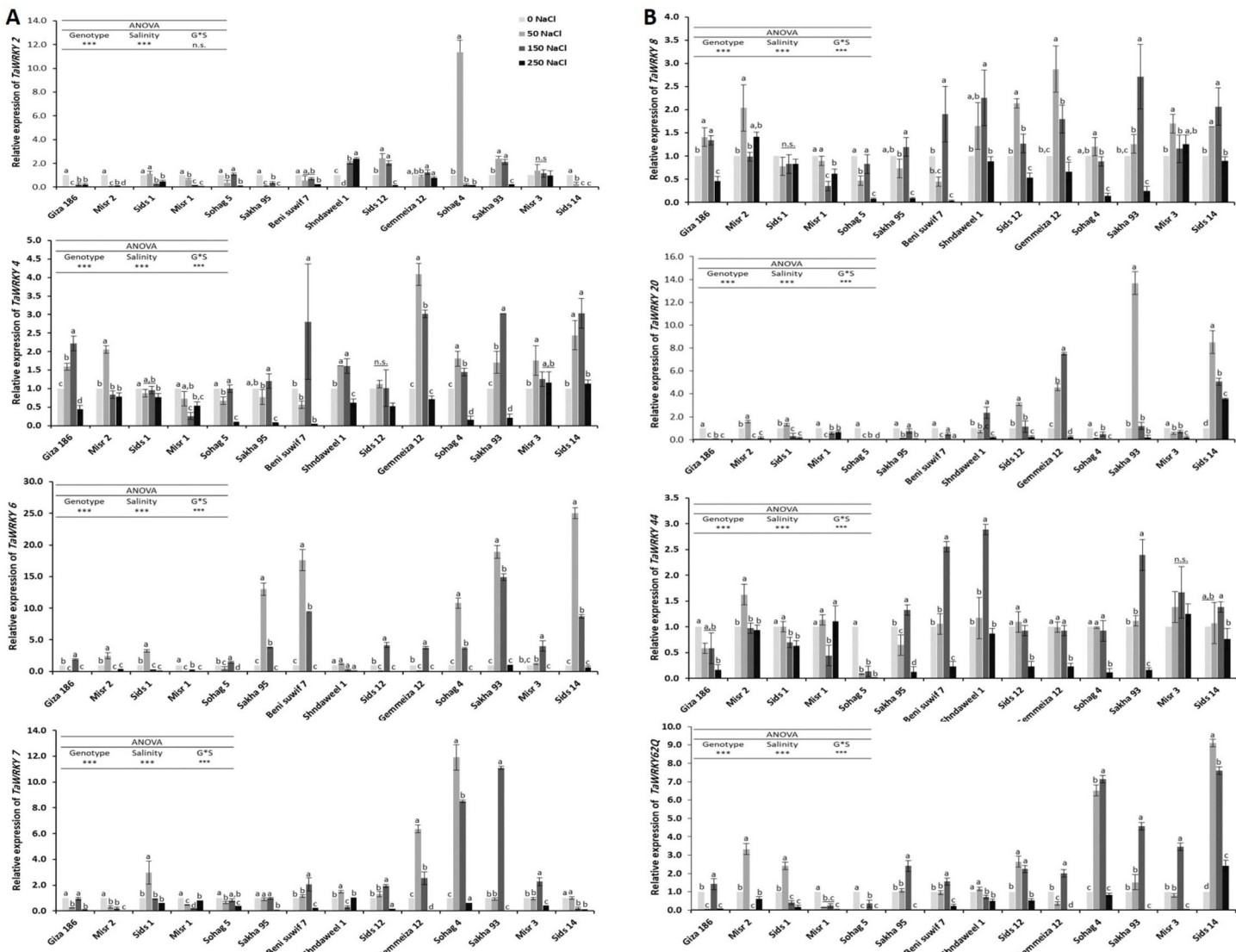

**Fig 1.** Expression patterns of (A) *TaWRKY2*, *4*, *6*, *7* and (B) *TaWRKY8*, *20*, *44*, *62Q* genes under NaCl treatments. The vertical ordinates are the relative expression levels (fold) compared with the non-stressed control. The horizontal ordinate is the treatment concentration at 0, 50, 150, and 250 mM NaCl. The expression level of *TaActin* was used as a loading control. Error bars represent standard deviations (SDs). All data are the means of three independent biological and technical replicates. ANOVA was performed to test the effects of salinity treatments and genotypes and their interaction. Post hoc analysis was performed using Tukey's test. Statistically significant differences are represented by different letters above the bars. Different lowercase letters indicate significant differences within salinity treatments for each genotype at $P < 0.05$.

150 mM NaCl, respectively), while Sids 14 had the highest number of upregulated genes at 250 mM NaCl (three genes).

Moreover, at 50 mM NaCl, Sohag 5 was the only genotype that did not show expression of any of the studied genes, while Misr1, Misr 2, and Sids 1 had the same trend at 150 mM NaCl; at 250 mM NaCl, Sids 14, Misr 3, Misr 1, Shandaweel 1, and Misr 2 were the only genotypes with upregulated expression for some studied genes. Overall, the expression levels of Sakha 93, Sids 12, and Shandaweel 1 showed an upregulation of in all the studied genes, followed by Gimmeza 12 and Misr 3 (seven genes), Sohag 4, Bani Suef 7, Sids 14 and Misr 2 (six genes), Sids 1 and Giza 168 (four genes), Sohag 5 (two genes), and finally Misr 1 (a slight increase in

expression of only *TaWRKY44)*. Based on these results, it was concluded that each gene had significant variability in its expression pattern between the contrasting wheat genotypes. The (S3 Fig) summarizes the observed gene expression patterns into seven groups according to their upregulation or downregulation after the NaCl treatments. In this regard, the expression of Sids 14 and Misr 3 for *TaWRKY 4*, Sohag 5, Giza 168, Sids 12, and Gimmeza 12 for *TaWRKY 6*, Shandaweel 1 for *TaWRKY7*, Shandaweel 1 for *TaWRKY2*, Misr 3, Sohag 4, Sakha 95, Bani Suef 7, Sakha 93, and Sids 14 for *TaWRKY 6*, Shandaweel 1, Sids 1, and Misr 2 for *TaWRKY 62Q*, and Sohag 5, Misr 1, Giza 168, Misr 3, Sohag 4, Sakha 95, and BaniSuef 7 for *TaWRKY20* are examples of the first pattern through the seventh.

### 3.3 Multivariate analysis

Based on gene expression of all studied genes, PCoA was performed to classify the studied genotypes and revealed eight significant axes (Fig 2A). The first axes explained 70.49, 66.15, and 54.98% of the total variance for 50, 150, and 250 mM NaCl, respectively. The results showed that the eigenvalue decreased with increasing salinity. Objects ordinated closer to one another were more similar than those ordinated further away. At 50 mM NaCl, all the

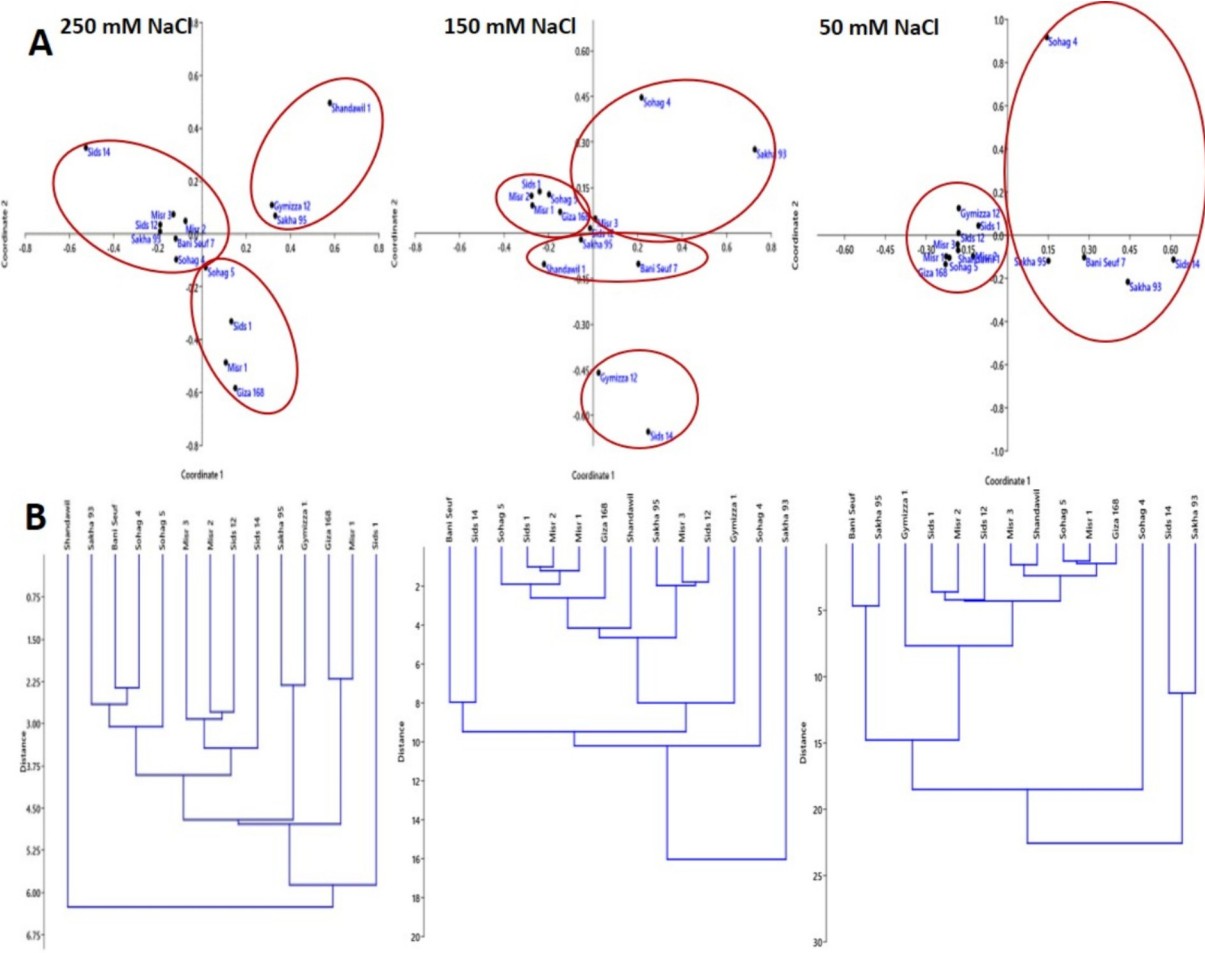

**Fig 2.** (A) Principal component analysis of *TaWRKY* gene expression for 14 genotypes under 50, 150, and 250 mM NaCl treatments. Every circle represents independent group genotypes according to their level of expression. (B) UPGMA-based dendrogram showing the genetic relationship among the 14 genotypes using gene expression results.

studied genotypes were grouped into four clusters: the first, second, and third groups included all the genotypes that had high or moderate levels of expression, while the fourth group included the rest of the genotypes that had low levels of expression. PCoA analysis at 150 mM NaCl revealed four different groups. Two of them included all the genotypes that were characterized by upregulated expression, while one group involved moderate expression and another included low expression. All the tolerant genotypes at 250 mM NaCl were in one separated group situated on the left side of the graph, while the other two groups were situated on the right side of the graph. The upper group included moderate resistance genotypes, while the lower group included sensitive genotypes. To support the PCoA results, hierarchical clustering analysis was also performed (Fig 2B), which produced similar groups and thus corroborated the PCoA results.

On the other hand, due to the importance of ISSR and SCoT data in distinguishing the studied genotypes, the relationships between the studied genes were further analysed to identify the specific potential genes and DNA bands that might be responsible for salinity tolerance in the studied genotypes. CCA was applied using markers and gene expression datasets under different treatments. The genetic relationship showed eight significant axes. The three major principal axes in the 50, 150, and 250 mM NaCl treatments accounted for 20.11%, 28.69%, 20.1% and 41%, 20.53%, 44.86% of the total variation for ISSR and SCoT, respectively. Among the studied genotypes, most of the detected bands were revealed by either SCoT or ISSR (especially the unique ones), and the highest expressed genes for every studied genotype were located together in the same groups (Fig 3A and 3B). For example, we found that *TaWRKY7* expression at 50 mM NaCl for Sohag 4 and its unique ISSR bands (H20, I9) grouped together, while *TaWRKY2* expression for Sohag4 and its unique SCoT bands (B8) grouped together. Moreover, at 150 mM NaCl, Sakha 93 was clustered with its unique ISSR bands (F9, H13, H17, and H21) and *TaWRKY 62Q*. Additionally, Sids 14 was clustered with its unique SCoT bands (B7, E8, and E12) and *TaWRKY6*. At 250 mM NaCl, Sids 14 and its unique ISSR bands (F11, G3, and H23) as well as the most highly expressed gene for Sids 14 and *TaWRKY 20* were located in the same cluster. In addition, SCoT unique bands (B7, E8, and E12) for Sids 14 and the expression of *TaWRKY 20* were located in the same clusters. These findings revealed that these molecular indicators are important parameters for assessing tolerance to abiotic stresses, including salinity. Generally, the results of DNA marker and gene expression analyses revealed high genetic diversity among all the studied genotypes. To confirm this variability, for subsequent experiments we selected some genotypes that showed high differences in genetic variability (Sids 14, Sohag 4, Sakha 93, Shandaweel 1, Gemmeiza 12, Misr 2, Misr 1, and Giza 168) to analyse their tolerance to salinity stress by studying biochemical, isozyme, and protein characteristics.

## 3.4 Impact of salinity on the photosynthetic pigments

Analysis of variance revealed significant variation among the studied genotypes under different salinity treatments and the interaction for Chl a, Chl b, and total carotenoid contents at $P \leq 0.05$. Tukey's test revealed significant differences among Sakha 93, Sohag 4, and Sids14 for Chl a, Sids 14 for Chl b, Sids 14, and Sohag 4 for total carotenoid content, and the rest of the genotypes. These results showed that the genotypes responded differently when they were subjected to salt conditions. Our results revealed an inverse relationship between salinity and photosynthetic pigment content. Moreover, we observed that the reduction percentage in the 150 mM NaCl treatment was higher than in the 50 mM NaCl treatment. When the NaCl concentration was increased to 150 mM NaCl in the growth medium, the Chl a, Chl b, and total carotenoid contents decreased to 78.9%, 60.1%, and 54.4% in Giza 168, respectively, compared with the control, while the Chl b and total carotenoid contents decreased to 17.8% and 3% in Sids

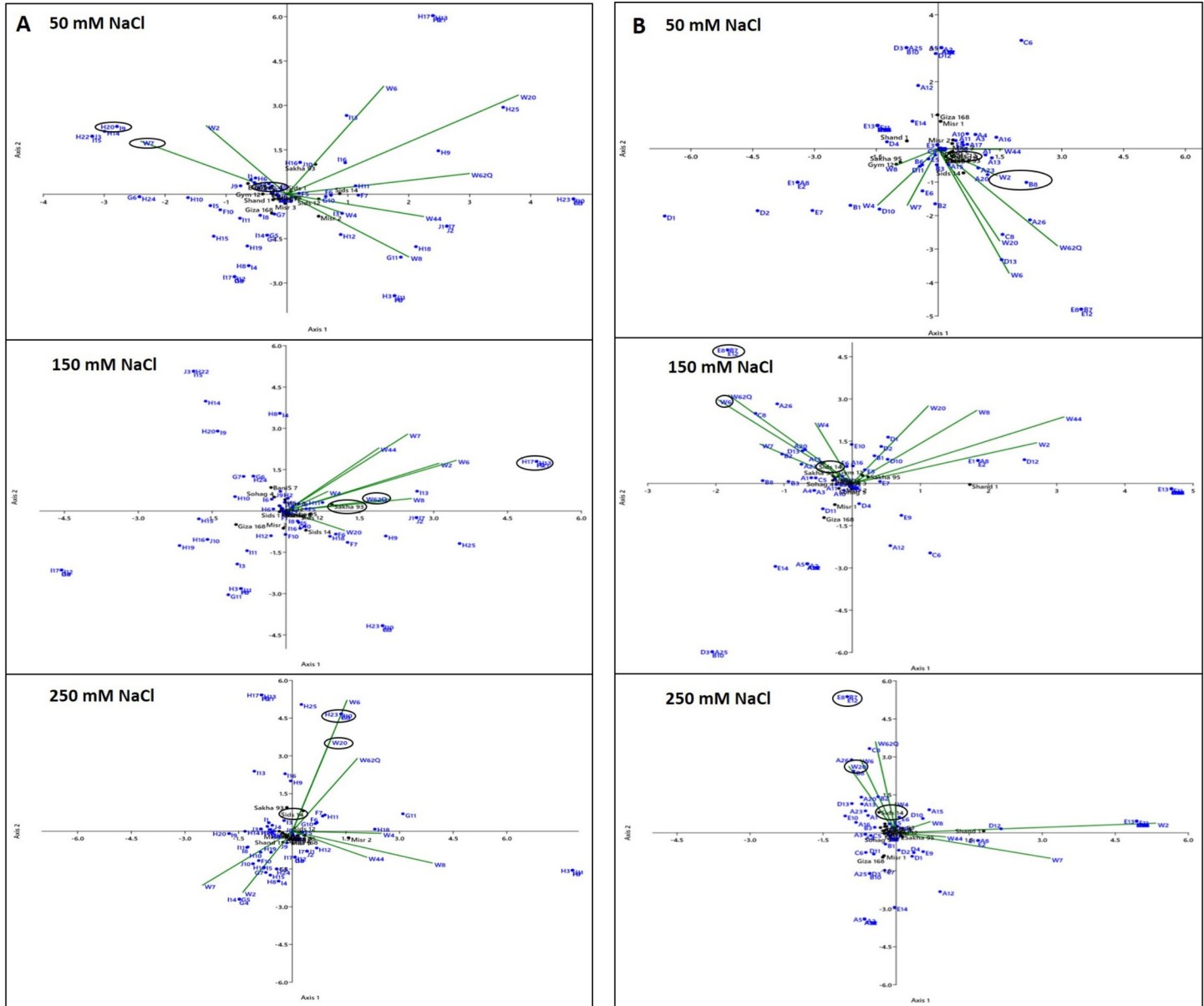

**Fig 3.** Canonical correspondence analysis (CCA) using (A) ISSR and (B) SCoT banding patterns and values of *TaWRKY* gene expression under 50, 150, and 250 mM NaCl for the studied genotypes. All primers and amplified bands with their molecular size were coded according to S1 File. Each gene is abbreviated as follows: *TaWRKY2* (W2), *TaWRKY4* (W4), *TaWRKY6* (W6), *TaWRKY7* (W7), *TaWRKY8* (W8), *TaWRKY20* (W20), *TaWRKY44* (W44), and *TaWRKY 62Q* (W62Q).

14 and Sohag 4, respectively. Interestingly, Chl a increased to 16% for Sakha 93 compared with the control. In the same context, Sakha 93, Sids 14, and Sohag 4 had the highest means at 150 mM NaCl for Chl a, Chl b, and total carotenoid contents, respectively. In contrast, the Giza 168, Misr 2, and Misr 1 genotypes had the lowest means at 150 mM NaCl (Fig 4A).

### 3.5 Impact of salinity on the proline, total sugar, and phenol contents

ANOVA results revealed highly significant variation between the salinity concentrations and genotypes at $P \leq 0.05$ for the proline, total sugar, and phenol content parameters, while the interaction between them was also highly significant for only the total sugar and phenol

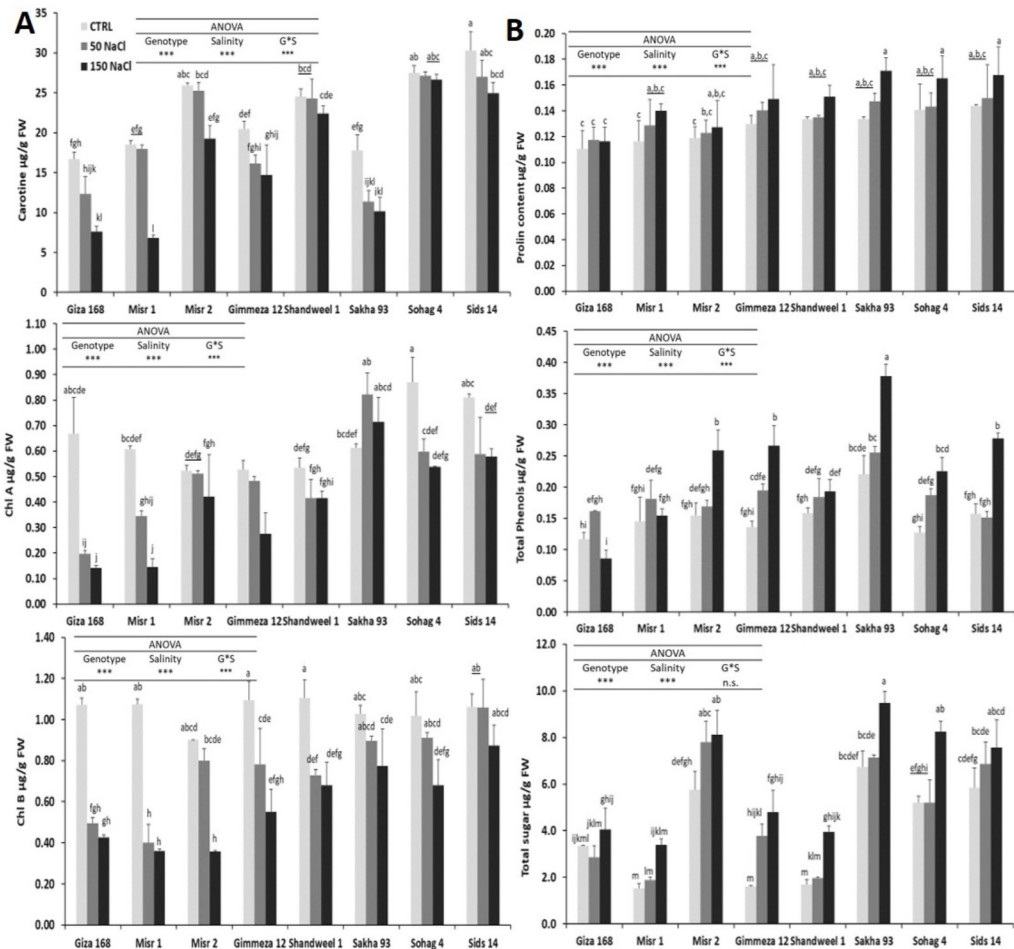

**Fig 4.** Biochemical traits: (A) chlorophyll a, chlorophyll b, and total carotenoid contents. (B) Proline, total phenolic, and sugar contents in wheat leaves under 0, 50, and 150 mM NaCl treatments. ANOVA was performed to test the effects of salinity treatments and genotypes and their interaction. The post hoc analysis was performed using Tukey's test. Statistically significant differences are represented by different letters above the bars. Different lowercase letters indicate significant differences among salinity treatments and genotype at $P < 0.05$. Values are the means of three observations, and error bars are standard deviations of the means.

contents, indicating the occurrence of differential genotype responses to salt stress. In this regard, there was a significant increase in the proline content in salt-treated leaves for Sids14, Sakha 93, and Sohag 4. Significant increases for Sakha 93 in the total sugar and phenol contents, compared to the other genotypes. In contrast, no increase in Giza 168 was observed at 150 mM NaCl for the total sugar and phenol contents. In this regard, the highest means of the proline, total sugar, and phenol contents at 150 mM NaCl concentrations were observed for Sakha 93, while Giza 168, Misr 1, and Giza 168 recorded the lowest values of proline, total sugar, and phenol, respectively (Fig 4B). The increasing percentage at 150 mM NaCl was higher than that at 50 mM NaCl. Consequently, the highest increasing percentages were 198% (Gemmeiza 12), 97% (Gemmeiza 12), and 21% (Sakha 93) for the total sugar, phenol, and proline contents, respectively, at 150 mM NaCl compared with the control, while the Giza 168 genotype recorded the lowest increasing percentages (21% and 5%) for the total sugar and proline contents, respectively. Interestingly, the total phenol content decreased to 27% for Giza 168 compared with the control.

## 3.6 Impact of salinity on isozyme profiles

Isozyme band intensities belonging to the selected genotypes were investigated. PX1 to PX5 isoforms for peroxidase and PPO1 to PPO5 isoforms for polyphenyl oxidase were clearly amplified for all studied genotypes/treatments (Fig 5A and 5B). According to the band intensities, the observed isoforms' expression levels were grouped into five different patterns. The first pattern included all the genotypes that had constant band intensities for all isoforms in the control and different salinity treatments. For example, PX1, 2, 3, 5 in Gemmeiza 12 and PX1, 3, 4 in Sohag 4, while PPO3, 4, 5 in Misr 1 and PPO1, 5 in Shandaweel 1 also belonged to this cluster. The second pattern involved all genotypes in which the isozyme activities decreased after 50 mM NaCl and increased after 150 mM NaCl treatments compared with the control. For example, Sids 14 (PX2, 3, 4, 5), Sakha 93 (PX3, 4, 5), Sohag 4 (PX1, 2, 3, 4, 5), and Sakha 93 (PPO1, 2) belonged to this group. In addition, the third pattern included all genotypes in which the band intensities increased after 50 mM NaCl and decreased after 150 mM NaCl treatments compared with the control. For example, PX3, 4, 5 isoforms for Misr 2 and PPO1, 2, 3 for Giza 168 belonged to this pattern. The fourth pattern

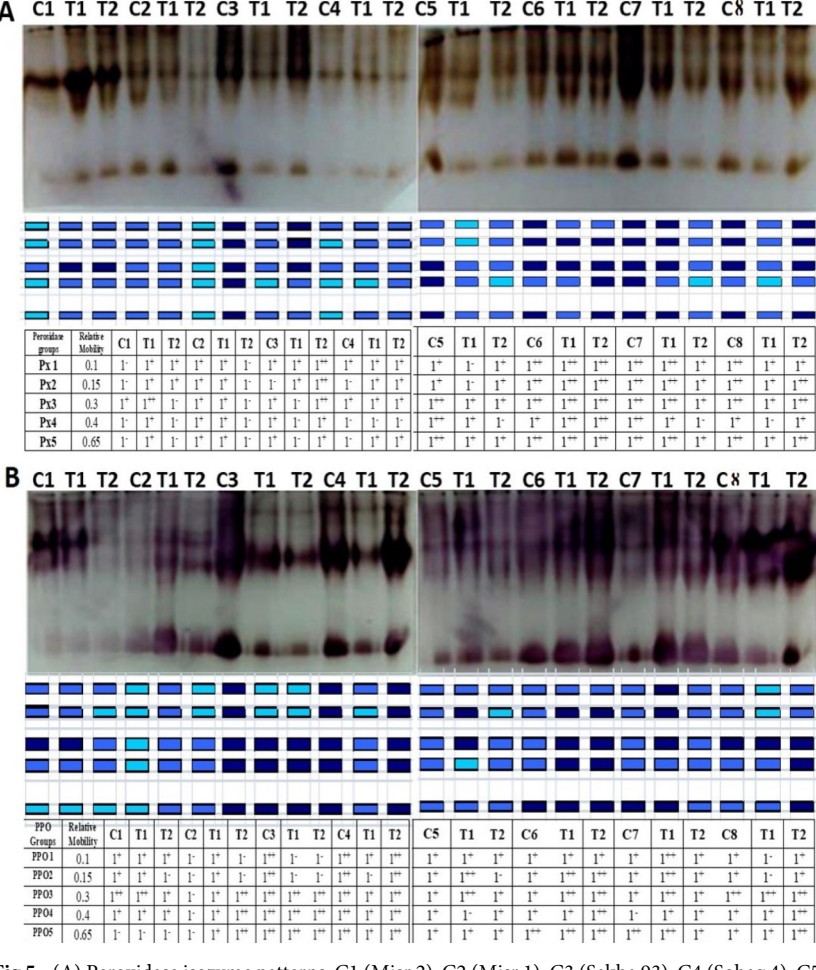

**Fig 5.** (A) Peroxidase isozyme patterns. C1 (Misr 2), C2 (Misr 1), C3 (Sakha 93), C4 (Sohag 4), C5 (Shandaweel 1), C6 (Gemmeiza 12), C7 (Giza 168), and C8 (Sids14). (B) polyphenyl oxidase isozyme patterns. C1 (Misr 2), C2 (Sids 14), C3 (Misr 1), C4 (Sohag 4), C5 (Shandaweel 1), C6 (Gemmeiza 12), C7 (Giza 168), and C8 (Sakha 93) under C (control), T1 (50), and T2 (150) mM NaCl treatments. ++ High-density band, +moderate-density band, and—low-density band.

involved all genotypes with band intensities that increased in both salinity treatments compared with the control. For example, the PX1, 2 isoforms in Sakha 93, the PX4 isoform in Gemmeiza 12 as well as PPO2, 3, 4 in Gemmeiza 12 and Sids 14, and PPO 4, 5 in Sakha 93 belonged to this cluster. The fifth pattern involved all the genotypes for which the band intensities gradually decreased. For example, Sids 12 (PX2), Sakha 95 (PX5), Giza 168 (PPO5), and Sakha 95 (PPO5) belonged to this pattern. PPO, and POX expression levels were clearly constitutively different between the control and salinity stress treatments for all the studied genotypes.

## 3.7 Impact of salinity on total protein content

SDS-PAGE was performed to characterize the protein patterns involved in the salinity response. A total of 18 polypeptides showed little heterogeneity among the control and salinity-treated genotypes, with molecular weights (MWs) ranging from 21 to 197 kDa. Compared with the control, most of the polypeptide bands were not affected by salinity. Notably, protein pattern analysis of Giza 168 and Misr 1 showed that the polypeptides with MWs ranging from 21–46, 74–109, and 158–193 kDa were prominent in the control and all the studied treatments. Additionally, Shandaweel 1 and Gemmeiza 12 were shared in polypeptides with MWs ranging from 22–26, 42–90, and 197 kDa. Moreover, polypeptide bands with MWs ranging from 24–87 kDa were commonly found in the Misr 2, Sakha 93, and Sids 14 genotypes. Conversely, salinity treatments resulted in the induction of a new band with an MW of 33 kDa at 150 mM NaCl for Gemmeiza 12, while other new bands appeared in seedlings treated with 50 and 150 NaCl in the Misr1, Giza 168, and Misr 2 genotypes with MWs of 56 kDa, 56, 116, and 133 kDa and 114 kDa, respectively. Additionally, the NaCl treatments resulted in the absence of specific bands that already appeared in the control treatment in Giza (134 and 61 kDa), Sohag 4 (32, and 29 kDa), and Misr1 (53 kDa) for T1 and T2, respectively, while this phenomenon was observed in Sakha 93 (114 and 121 kDa) for T2. From Table 3, we concluded that (Sids 14, Sakha 93), (Sids 14, Sakha 93, Giza168, and Misr 2) and (Sids 14, Giza 168, and Misr 2) scored the highest amplified bands under the control, T1, and T2 conditions, respectively. In addition, the highest polymorphism percentages were 15.4% under control conditions for the Sohag 4 and Giza 168 genotypes. Moreover, under T1 and T2 conditions, Giza 168 recorded the highest polymorphism percentage (21.4%). Moreover, Sohag 4, Giza 168, and Gemmeiza 12 produced four and one unique bands under normal and T2 conditions, respectively.

**Table 3.  Number and types of amplified protein bands and polymorphism percentages generated under different salinity treatments.**

|  | TAB | | | MB | | | PB | | | UB | | | %P | | |
|---|---|---|---|---|---|---|---|---|---|---|---|---|---|---|---|
|  | C | T1 | T2 | C | T1 | T2 | C | T1 | T2 | C | T1 | T2 | C | T1 | T2 |
| Sids 14 | 14 | 14 | 14 | 14 | 14 | 14 | 0 | 0 | 0 | 0 | 0 | 0 | 0 | 0 | 0 |
| Gemmeiza 12 | 9 | 9 | 10 | 9 | 9 | 9 | 0 | 0 | 1 | 0 | 0 | 1 | 0 | 0 | 10 |
| Sohag 4 | 13 | 11 | 11 | 11 | 11 | 11 | 2 | 0 | 0 | 2 | 0 | 0 | 15.4 | 0 | 0 |
| Sakha 93 | 14 | 14 | 12 | 12 | 12 | 12 | 2 | 2 | 0 | 0 | 0 | 0 | 14.3 | 14.3 | 0 |
| Shandaweel 1 | 9 | 9 | 9 | 9 | 9 | 9 | 0 | 0 | 0 | 0 | 0 | 0 | 0 | 0 | 0 |
| Giza 168 | 13 | 14 | 14 | 11 | 11 | 11 | 2 | 3 | 3 | 2 | 0 | 0 | 15.4 | 21.4 | 21.4 |
| Misr 2 | 13 | 14 | 14 | 13 | 13 | 13 | 0 | 1 | 1 | 0 | 0 | 0 | 0 | 7.1 | 7.1 |
| Total | 85 | 85 | 85 | 79 | 79 | 79 | 6 | 6 | 5 | 4 | 0 | 1 | 7.1 | 7.1 | 6 |

**TAB:** total amplified bands, **MB:** monomorphic band, **PB:** polymorphic band, **UB:** unique band, **%P:** Percent of polymorphism.

## 4 Discussion

To fully understand the negative effect of elevated salinity in wheat, multiple organizational analyses at the molecular and biochemical levels were performed. Genetic diversity is a requirement for developing salt-tolerant wheat varieties [45]. Accordingly, we performed ISSR and SCoT analyses to determine their elevated reproducibility and great power in polymorphism discovery in wheat [11]. Analysis of the amplification patterns of the studied genotypes showed no difference in the total number of generated bands. In fact, ISSR markers were more efficient than SCoT markers regarding polymorphism detection since they detected 86% compared to 82% polymorphisms. These results are consistent with a previous report [46].

One possible explanation for such an effect of the total of 66 polymorphic alleles and the average of 13.2 alleles per ISSR primer observed in this study could be justified by the ISSR markers' nature. ISSR markers amplify the inter-microsatellites region and have the advantage of being more polymorphic [47, 48]. The detected high level of polymorphism was indicative of the greater genetic diversity among the tested genotypes, which can be effectively utilized for gene tagging and genome mapping of crosses to introgress the favourable traits into the cultivated genotypes.

To determine the efficiency of the studied markers in discriminating and establishing genetic relationships among different wheat genotypes, various crucial parameters in the study of genetic diversity in plant species [49, 50] were estimated. The average of EMR (11.5) and MI (8.63) for the ISSR markers were higher than those of the SCoT markers 10.67 and 7.80, respectively. In contrast, the average Rp value was higher for SCoT markers (9.15) than ISSR markers (8.52), which indicated a strong ability of the SCoT markers to reflect the genetic diversity. In the same context, the average PIC value for SCoT markers (0.71) was higher than ISSR (0.69), suggesting a good capacity of the SCoT system to present the polymorphic level in the investigated genotypes. Similar results were revealed by [51]. Furthermore, the higher number of alleles per locus and number of effective alleles, the more polymorphic was the population and reflected the size of the population variation. In this regard, our results showed that the mean of the observed number of alleles (1.79, 1.81) was less than (Na) the means reported by [52] and higher than the means reported by [53] using SCoT and ISSR, respectively. Using ISSR markers, [54] found that the average of (Ne) was (1.63) for *Pistacia Vera*, which was higher than the values obtained in the present study (1.25, 1.24) for ISSR and SCoT, respectively. The expected heterozygosity (He) value was used to measure the genetic diversity of a population. Overall, the higher the value of (He), the lower is the genetic consistency of the revealed population, and the richer the genetic diversity of the population is [52]. In this research, the average value of (He) using the ISSR or SCoT primers was (0.15), which is comparable to the value (0.16) reported by [55] for *Salvadora persica* genotypes. In addition, this value was less than the average detected by [52] (0.69) using the SCoT marker. Moreover, ISSR showed its effectiveness in discriminating the tested genotypes by generating more unique bands than SCoT. These bands could be recognized as markers correlated with salinity tolerance in wheat. Similarly, Metwali and Almaghrabi [56] showed a specific product for cultivar Sakha 93 with sizes of approximately 550 and 700 bp, and a specific fragment was detected in Sids12 at 350 bp. Overall, both marker types demonstrated high values for genetic diversity parameters and the total number of polymorphic loci, with higher values obtained for ISSR markers in comparison to SCoT. The latter point revealed that the ISSR marker was more efficient in estimating genetic variation, which is in accordance with [11].

To confirm the genetic variability among the studied genotypes, a transcription analysis of TF *TaWRKY* genes was performed under salinity stress. The variations in gene expression among genotypes were less at low salinity concentrations than at high concentrations, suggesting that the

selection criteria could be considered convenient for screening wheat cultivars only when they were estimated under high salinity concentrations. This result is in line with the findings of [57].

Expression analysis demonstrated that all the studied genes were induced by the presence of NaCl. In addition, *TaWRKY4*, *8*, *6* had the highest numbers of upregulated genotypes. Our findings are consistent with previous results reporting the expression levels of *TaWRKY44* and *TaWRKY4* [58], *TaWRKY93* [59], and *TaWRKY7* and *TaWRKY12* [16] in plants exposed to salinity. In contrast, *TaWRKY2*, *20* had the lowest numbers of upregulated genotypes. Similarly, *TaWRKY34* did not respond within 1 h of stress, and over 12 h post-stress, there was no expression of *TaWRKY20* [16]. The quick upregulation of *TaWRKY6* expression levels in Sids 14, Sakha 93, Bani Suef 7, and Sakha 95 after various stress treatments indicated a remarkable role at the initial stages of the stress response (S3 Fig). Strongly suggests that WRKY gene expression in the studied genotypes was somewhat complicated due to the use of different salinity concentrations and genotypes. In fact, the expression level may change during exposure to stress, with upregulation followed by downregulation or vice versa [60].

Gene expression results reflected a high genetic diversity in the studied collection, which was confirmed by PCoA to assist in gathering the manners in which gene expression responses varied in contrasting cultivars.

On the other hand, the association of molecular markers with gene expression is an important factor to understand the genetic role of tolerance by predicting the genomic regions that affect the plant response. Therefore, the results of CCA analysis showed that the distribution pattern of the genotypes was mainly correlated with their upregulated genes. The clear correlation between most of the unique bands of every studied genotype and the expression levels of the studied genes suggested that these bands could serve as selectable markers for salinity tolerance in wheat; however, further studies aimed at purification, sequencing and analysis of these bands might be necessary in future work. In this regard, ISSR has been used to recognize markers correlated with salinity tolerance in wheat [61]. Remarkably, the SCoT marker is produced from the functional region of the genome, while the amendment of gene expression by WRKY proteins first occurs through DNA binding at certain cis-regulatory elements, the W-box elements, which are short sequences located in the promoter region of particular genes [62]. These findings might suggest that the detected unique bands revealed by SCoT were responsible for *TaWRKY* expression in the studied genotypes.

Analysis of the studied biochemical traits clearly showed that the photosynthetic pigments were significantly reduced, while the proline, total sugar, and total phenol contents were significantly increased under salinity stress in the suggested tolerant and sensitive genotypes. The noted chlorophyll degradation is consistent with results obtained by [63]. In the same context, Abd Elhamid *et al*. [22] stated that Sakha 93 and Giza 168 recorded the highest and lowest values of total phenolic content in response to salinity stress, which is in agreement with our findings. Furthermore, significant increases in the proline and soluble sugar contents of wheat genotypes have been noted by [21] and [64] after salinity stress, respectively. In contrast, Wang and Nil [65] showed that the chlorophyll content increased under salt conditions in *Amaranthus*, while Lutts *et al*. 1999 [66] showed a negative correlation among the proline and sugar content accumulation and salinity tolerance.

In the current study, as salinity levels rose, some PX and PPO isoforms increased significantly. This increase in the leaves was more conspicuous in Sakha 93, Gemmeiza 12, Shandaweel 1, Sohag 4, and Sids 14 than in Misr 2, Misr 1, and Giza 168, suggesting that tolerant cultivars have the best $O_2$ radical scavenging ability. However, this increase is a strategy for improving salt tolerance, and such upregulation often occurs to defend protein systems after stress. These results are consistent with [67], who pointed out that salt stress increased the peroxidase band intensity. Consistently, Abdelsalam and Kandil [68] showed that the Sakha 93

genotype had high enzyme activity under NaCl. On the other hand, band intensities for some other isoforms were higher in the salt-sensitive than the salt-tolerant cultivars compared with the control plants, thereby enabling plants to protect themselves against oxidative stress. Similarly, Meratan *et al*. [69] reported that POX isozymes showed declining and increasing patterns in *A. sordidum* and *A. glandulosum* under high salt concentrations, respectively. These inconsistent results may be because ROS accumulation and the upregulation of antioxidant enzymes are dependent on the plant species, plant genotype, stress severity, stress duration, plant development, and metabolism [70].

Likewise, salinity treatments caused either a decrease or increase in the level of total proteins or caused some proteins to completely disappear compared with the control. In this regard, Giza 168 and Sakha 93 were characterized by the appearance and absence of some protein bands compared with the control under salinity treatment. Generally, salt-tolerant and salt-sensitive genotypes did not vary significantly in leaf proteins, in agreement with [63]. Consequently, salt tolerance is not constantly related to protein production and depends on the nature of the plant species [71].

Briefly, according to molecular and biochemical results obtained in the present study, genotypes Sids 14, Misr 3, Sakha 93, Sohag 4, Gemmeiza 12, and Sids 12 were clearly ranked as tolerant genotypes. Thus, these salt-tolerant genotypes could be beneficial as genetic resources for future breeding programs, thus expanding the genetic base for salt tolerance breeding in wheat. In addition, Shandaweel 1, Bani Seuf 7, Sakha 95, and Misr 2 were ranked as moderate genotypes, while Sids 1, Giza 168, Misr 1, and Sohag 5 were classified as sensitive genotypes. These results support the findings of several authors, such as [68, 72, 73], who ranked genotype Sakha 93 as one of the most tolerant wheat cultivars to salinity based on various physiological and growth parameters. However, [56, 63] classified Giza 168 as salt susceptible.

## 5 Conclusion

The productivity of wheat can be improved using tolerant genotypes [74]. However, evaluation of the salt tolerance potential in wheat through marker and stress-related gene expression analysis could potentially diminish the cost of breeding programmes and be a powerful strategy for selection of the most salt-tolerant genotype. ISSR markers were more efficient and had a stronger discriminating power than SCoT markers for the studied wheat genotypes, as indicated by the high values of the genetic diversity indices, polymorphism percentage, and number of specific bands. These bands might be considered useful markers linked to salinity tolerance in wheat breeding programmes. The maximum number of unique bands was scored by primers (HB-10, SCoT 2, and SCoT 8), while Shandaweel 1 had the highest number of unique bands detected by both markers. In the same context, the differential expression of eight *TaWRKY* genes was shown in leaf tissues of the Egyptian salinity-tolerant and salinity-susceptible genotypes for the first time. The highest and lowest expression levels were observed for Sids 14 and Misr 1 across all the studied concentrations of NaCl. At the biochemical level, it was noted that for most of the studied traits, the Sakha 93 and Giza168 genotypes achieved the highest and lowest values. In summary, salt stress clearly has detrimental effects on the molecular and biochemical processes associated with salinity tolerance.

## Supporting information

**S1 Fig.** (A) SCoT and (B) ISSR fingerprinting of wheat genotypes: M; DNA marker, lanes 1–14; Misr-2, Misr-3, Sids-1, Sids-12, Sids-14, Sakha-93, Bani Suef-7, Sohag-4, Sohag-5, Shandweel-1, Sakha-95, Gemmeiza 12, Giza-168 and Misr-1.
(TIF)

**S2 Fig. Phylogenetic relationships detected by cluster analysis using ISSR and SCoT, combined data among the studied genotypes.** (1) Misr 2, (2) Misr 3, (3) Sids 1, (4) Sids 12, (5) Sids 14, (6) Sakha 93, (7) Bani Seuf 7, (8) Sohag 4, (9) Sohag 5, (10) Shandaweel1, (11) Sakha 95, (12) Gemmeiza 12, (13) Giza 168, and (14) Misr 1. Dendrogram calculated using Jaccard's similarity coefficients and the UPGMA algorithm.
(TIF)

**S3 Fig. Summary of different expression patterns for all studied genes and genotypes.** (↑) Means upregulation, (↓) means downregulation. The first pattern is coloured red and involves all the genotypes in which its expression was upregulated for all studied genes at 50, 150, and 250 mM NaCl. The second pattern is coloured orange and includes all the genotypes in which expression decreased after being subjected to 50 mM NaCl and thereafter increased after being exposed to 150 mM NaCl, ultimately being downregulated after exposure to 250 mM NaCl. The third pattern is coloured blue and includes all the genotypes; its expression was increased after exposure to 50 mM NaCl and then downregulated after exposure to 150 mM NaCl, finally being upregulated following exposure to 250 mM NaCl. The fourth pattern is coloured light grey and includes all the genotypes in which its expression decreased after exposure to 50 mM NaCl and thereafter increased after exposure to 150 and 250 mM NaCl. The fifth pattern is coloured yellow and includes all the genotypes in which its expression increased after exposure to 50 and 150 mM NaCl and then decreased after exposure to 250 mM NaCl. The sixth pattern is coloured dark grey and includes all the genotypes in which its expression increased after exposure to 50 mM NaCl and then decreased after exposure to 150 and 250 mM NaCl. The seventh pattern is coloured green and includes all the genotypes in which its expression was downregulated for the studied genes at 50, 150, and 250 mM NaCl.
(TIF)

**S1 Table. List of the primer names and their nucleotide sequences used in the study for the ISSR and SCoT procedure.**
(DOCX)

**S2 Table. Sequences of the primers used in the real-time PCR.**
(DOCX)

**S3 Table. The similarity matrix based on ISSR and SCoT data.**
(DOCX)

**S4 Table. Total number of genotypes/induced genes under different NaCl concentrations.**
(DOCX)

**S5 Table. Total number of upregulated genes/genotypes under different NaCl concentrations.**
(DOCX)

**S1 Raw image.**
(PDF)

**S1 File.**
(XLSX)

## Author Contributions

**Conceptualization:** Diaa Abd El-Moneim.

**Data curation:** Diaa Abd El-Moneim.

**Formal analysis:** Salah M. H. Gowayed, Diaa Abd El-Moneim.

**Funding acquisition:** Salah M. H. Gowayed.

**Investigation:** Diaa Abd El-Moneim.

**Methodology:** Salah M. H. Gowayed.

**Project administration:** Salah M. H. Gowayed.

**Resources:** Diaa Abd El-Moneim.

**Validation:** Diaa Abd El-Moneim.

**Visualization:** Salah M. H. Gowayed.

**Writing – original draft:** Salah M. H. Gowayed, Diaa Abd El-Moneim.

**Writing – review & editing:** Salah M. H. Gowayed, Diaa Abd El-Moneim.

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
