## [Decision Letter · Decision Letter 0]

30 Dec 2020

PONE-D-20-32358

Detection of genetic divergence among some wheat (Triticum aestivum L.) genotypes using molecular and biochemical indicators under salinity stress.

PLOS ONE

Dear Dr. Abd El Moneim,

Thank you for submitting your manuscript to PLOS ONE. After careful consideration, we feel that it has merit but does not fully meet PLOS ONE’s publication criteria as it currently stands. Therefore, we invite you to submit a revised version of the manuscript that addresses the points raised during the review process.

We look forward to receiving your revised manuscript.

Kind regards,

Mehdi Rahimi, Ph.D.

Academic Editor

PLOS ONE

Journal Requirements:

Reviewers' comments:

Reviewer's Responses to Questions

**Comments to the Author**

1. Is the manuscript technically sound, and do the data support the conclusions?

Reviewer #1: Yes

Reviewer #2: Yes

2. Has the statistical analysis been performed appropriately and rigorously? 

Reviewer #1: Yes

Reviewer #2: Yes

3. Have the authors made all data underlying the findings in their manuscript fully available?

Reviewer #1: Yes

Reviewer #2: Yes

4. Is the manuscript presented in an intelligible fashion and written in standard English?

Reviewer #1: Yes

Reviewer #2: Yes

5. Review Comments to the Author

Reviewer #1: Wheat is an important crop and staple in many countries. Thus, any restrictions to its yield or production through salinity stress is of high concern. This paper describes the molecular and biochemical diversity in response to salt stress. The manuscript has been well written, with a few issues I would like to bring the authors attention to.

Firstly, the formatting of the paper should be checked to ensure that they meet the requirements of the journal, the major issue being the tables and the headings. (Kindly refer to the author guidelines for that).

Secondly, mostly seen at the materials and methods section, references were made to numbers, which I believe is not in right order. For example, "seeds were sterilized and then germinated for three weeks using the method described by [28]". Please rectify.

Thirdly, parameters describing the informativeness of the markers including average number of alleles per locus, average number of effective alleles, the mean heterozygosity per locus could be included.

Fourthly, the discussion section is all put together, with no paragraphing makes reading difficult. Moreover, the explanation could be concise and better explaining.

Finally, authors should scan through the entire manuscript for minor typographical and grammatical errors, as such.

Reviewer #2: The paper is written in very well manner. Still need improvement in english. Overall, the paper is good and can be proceed further for evaluation. Stats is well explained and need no further addition

6. PLOS authors have the option to publish the peer review history of their article (what does this mean?). If published, this will include your full peer review and any attached files.

Reviewer #1: No

Reviewer #2: **Yes: **Rabail Afzal

---

## [Author Response · Author response to Decision Letter 0]

25 Jan 2021

Response to Reviewer 1 Comments

First, we are very appreciative of the great efforts of the reviewer in reviewing our manuscript, and we are very thankful for all the valuable comments and suggestions, which have improved our manuscript. In this regard, below we provide detailed answers to each of the requests and concerns raised by the reviewers. Moreover, we have attached a revised version of our manuscript that includes all the suggested corrections.

Point 1: Firstly, the formatting of the paper should be checked to ensure that they meet the requirements of the journal, the major issue being the tables and the headings.

Response 1: We have checked the formatting of the paper, and the revised manuscript as well as the tables and the figures follows the journal instructions.

Point 2: Secondly, mostly seen at the materials and methods section, references were made to numbers, which I believe is not in right order. For example, "seeds were sterilized and then germinated for three weeks using the method described by [28]". Please rectify.

Response 2: This has been corrected in the new revised manuscript file. Moreover, we have examined the entire manuscript to ensure that all the inserted references are numbered in the appropriate order.

Point 3: Thirdly, parameters describing the informativeness of the markers including average number of alleles per locus, average number of effective alleles, the mean heterozygosity per locus could be included.

Response 3: 

- All the required parameters have been calculated, and the results are included in Table 1. 

- Please note that we have reconstructed Table 1, which was two separate tables in the old version. In its new format, this table now includes all the estimated and calculated parameters from the old version of the manuscript (MB: monomorphic band, UB: unique band, PB: polymorphic band, TAB: total amplified bands, FS: fragment size, PIC: polymorphic information content, EMR: effective multiplex ratio, MI: marker index, P%: percent of polymorphism, Rp: resolving power) as well as all the new calculated parameters (Na: average number of alleles per loci, Ne: average number of effective alleles per loci, He: expected heterozygosity, uHe: unbiased expected heterozygosity). 

- In addition, please note that it was difficult to calculate the mean heterozygosity per locus for the studied ISSR and SCoT primers. Because the amplification bands are often dominant markers, it is impossible to effectively identify homozygotes and heterozygotes (Isabel et al., 1999; McGregor et al., 2000 and Burgess 2001). Therefore, we calculated the expected heterozygosity and unbiased expected heterozygosity.

- Additionally, in the Material and Methods section, we have included the methods that were used to calculate the new parameters. The Results and Discussion have been updated accordingly.

Point 4: Fourthly, the discussion section is all put together, with no paragraphing makes reading difficult. Moreover, the explanation could be concise and better explaining.

Response 4: This has been corrected in the revised manuscript. Please note that the format of the discussion has been reconstructed in separate paragraphs as requested. Simultaneously, we have examined the entire Discussion and deleted some parts, in addition to revising others for clarity and concision.

Point 5: Finally, authors should scan through the entire manuscript for minor typographical and grammatical errors, as such.

Response 5: Before we submitted our paper for review by your respected journal, we had our manuscript edited for proper English language, grammar, punctuation, spelling, and overall style by one or more of the highly qualified native English speaking editors at AJE (please see the attached certificate). In addition, we have reedited the corrected manuscript using the same service (please see the attached certificate). 

Response to Reviewer 2 Comments

Point 1: The paper is written in very well manner. Still need improvement in English. Overall, the paper is good and can be proceed further for evaluation. 

Response 1: Before we submitted our paper for review by your respected journal, we had our manuscript edited for proper English language, grammar, punctuation, spelling, and overall style by one or more of the highly qualified native English speaking editors at AJE (please see the attached certificate). In addition, we have reedited the corrected manuscript using the same service (please see the attached certificate).

---

## [Decision Letter · Decision Letter 1]

2 Mar 2021

PONE-D-20-32358R1

Detection of genetic divergence among some wheat (Triticum aestivum L.) genotypes using molecular and biochemical indicators under salinity stress.

PLOS ONE

Dear Dr. Abd El Moneim,

Thank you for submitting your manuscript to PLOS ONE. After careful consideration, we feel that it has merit but does not fully meet PLOS ONE’s publication criteria as it currently stands. Therefore, we invite you to submit a revised version of the manuscript that addresses the points raised during the review process.

We look forward to receiving your revised manuscript.

Kind regards,

Mehdi Rahimi, Ph.D.

Academic Editor

PLOS ONE

Journal Requirements:

Reviewers' comments:

Reviewer's Responses to Questions

**Comments to the Author**

1. If the authors have adequately addressed your comments raised in a previous round of review and you feel that this manuscript is now acceptable for publication, you may indicate that here to bypass the “Comments to the Author” section, enter your conflict of interest statement in the “Confidential to Editor” section, and submit your "Accept" recommendation.

Reviewer #1: All comments have been addressed

Reviewer #2: All comments have been addressed

2. Is the manuscript technically sound, and do the data support the conclusions?

Reviewer #1: Yes

Reviewer #2: Yes

3. Has the statistical analysis been performed appropriately and rigorously? 

Reviewer #1: Yes

Reviewer #2: Yes

4. Have the authors made all data underlying the findings in their manuscript fully available?

Reviewer #1: Yes

Reviewer #2: Yes

5. Is the manuscript presented in an intelligible fashion and written in standard English?

Reviewer #1: Yes

Reviewer #2: Yes

6. Review Comments to the Author

Reviewer #1: Manuscript is well written. However, authors should go through the formatting very well to ensure that they are consistent. Especially with the headings and sub-headings. In all, paper has been properly presented.

Reviewer #2: Please check and review this paper again. Some grammatical mistakes are still present. Some minor mistakes of grammer

7. PLOS authors have the option to publish the peer review history of their article (what does this mean?). If published, this will include your full peer review and any attached files.

Reviewer #1: No

Reviewer #2: **Yes: **Rabail Afzal

---

## [Author Response · Author response to Decision Letter 1]

3 Mar 2021

First, we are very appreciative of the great efforts of the reviewer in reviewing our manuscript, and we are very thankful for all the valuable comments and suggestions, which have improved our manuscript. In this regard, below we provide detailed answers to each of the requests and concerns raised by the reviewers. Moreover, we have attached a revised version of our manuscript that includes all the suggested corrections.

Response to Reviewer 1 Comments

Point 1: Manuscript is well written. However, authors should go through the formatting very well to ensure that they are consistent. Especially with the headings and sub-headings. In all, paper has been properly presented.

Response 1: thank you so much for your positive comment, we have checked the formatting of the paper, as well as the headings and sub-headings to be sure that all are in the suitable font. 

Response to Reviewer 2 Comments

Point 1: Please check and review this paper again. Some grammatical mistakes are still present. Some minor mistakes of grammar. 

Response 1: we have checked the entire manuscript to detect any grammatical mistakes. Please see the attached revised version that include all the corrections that we did.

---

## [Decision Letter · Decision Letter 2]

8 Mar 2021

Detection of genetic divergence among some wheat (Triticum aestivum L.) genotypes using molecular and biochemical indicators under salinity stress.

PONE-D-20-32358R2

Dear Dr. Abd El Moneim,

We’re pleased to inform you that your manuscript has been judged scientifically suitable for publication and will be formally accepted for publication once it meets all outstanding technical requirements.

Kind regards,

Mehdi Rahimi, Ph.D.

Academic Editor

PLOS ONE

Additional Editor Comments (optional):

Reviewers' comments:

Reviewer's Responses to Questions

**Comments to the Author**

1. If the authors have adequately addressed your comments raised in a previous round of review and you feel that this manuscript is now acceptable for publication, you may indicate that here to bypass the “Comments to the Author” section, enter your conflict of interest statement in the “Confidential to Editor” section, and submit your "Accept" recommendation.

Reviewer #2: All comments have been addressed

2. Is the manuscript technically sound, and do the data support the conclusions?

Reviewer #2: Yes

3. Has the statistical analysis been performed appropriately and rigorously? 

Reviewer #2: Yes

4. Have the authors made all data underlying the findings in their manuscript fully available?

Reviewer #2: Yes

5. Is the manuscript presented in an intelligible fashion and written in standard English?

Reviewer #2: Yes

6. Review Comments to the Author

Reviewer #2: All the comments and recommendations have been addressed in this revision. No more changes needded now

7. PLOS authors have the option to publish the peer review history of their article (what does this mean?). If published, this will include your full peer review and any attached files.

Reviewer #2: **Yes: **Rabail Afzal

---

## [Editor Report · Acceptance letter]

18 Mar 2021

PONE-D-20-32358R2 

Detection of genetic divergence among some wheat (*Triticum aestivum* L.) genotypes using molecular and biochemical indicators under salinity stress 

Dear Dr. Abd El-Moneim:

I'm pleased to inform you that your manuscript has been deemed suitable for publication in PLOS ONE. Congratulations! Your manuscript is now with our production department. 

Kind regards, 

on behalf of

Dr. Mehdi Rahimi 

Academic Editor

PLOS ONE